# Demystify Mamba in Vision: A Linear Attention Perspective

**Dongchen Han**[1]    **Ziyi Wang**[1]    **Zhuofan Xia**[1]    **Yizeng Han**[1]    **Yifan Pu**[1]    **Chunjiang Ge**[1]
**Jun Song**[2]    **Shiji Song**[1]    **Bo Zheng**[2]    **Gao Huang**[1*]

[1] Tsinghua University        [2] Alibaba Group

## Abstract

Mamba is an effective state space model with linear computation complexity. It has recently shown impressive efficiency in dealing with high-resolution inputs across various vision tasks. In this paper, we reveal that the powerful Mamba model shares surprising similarities with linear attention Transformer, which typically underperform conventional Transformer in practice. By exploring the similarities and disparities between the effective Mamba and subpar linear attention Transformer, we provide comprehensive analyses to demystify the key factors behind Mamba's success. Specifically, we reformulate the selective state space model and linear attention within a unified formulation, rephrasing Mamba as a variant of linear attention Transformer with six major distinctions: input gate, forget gate, shortcut, no attention normalization, single-head, and modified block design. For each design, we meticulously analyze its pros and cons, and empirically evaluate its impact on model performance in vision tasks. Interestingly, the results highlight the forget gate and block design as the core contributors to Mamba's success, while the other four designs are less crucial. Based on these findings, we propose a *Mamba-Inspired Linear Attention (MILA)* model by incorporating the merits of these two key designs into linear attention. The resulting model outperforms various vision Mamba models in both image classification and high-resolution dense prediction tasks, while enjoying parallelizable computation and fast inference speed. Code is available at https://github.com/LeapLabTHU/MLLA.

## 1    Introduction

Recently, state space models, exemplified by Mamba, have rapidly gained wide research interest. In contrast to the quadratic complexity of prevailing Transformer models, the state-space-based Mamba offers effective sequence modeling with linear complexity. This crucial property allows Mamba to handle extremely long sequences with manageable computational costs, making it a promising architecture for both natural language processing [14, 29] and visual recognition [57, 31].

However, Mamba is not the first model to achieve global modeling with linear complexity. Linear attention [26], an early work, was proposed as an computationally efficient alternative to the widely adopted Softmax attention [42], namely dot-product attention. Specifically, linear attention replaces the non-linear Softmax function in attention operation with linear normalization. This enables a change in computation order from $(QK^\top)V$ to $Q(K^\top V)$, thus reducing computation complexity from $\mathcal{O}(N^2)$ to $\mathcal{O}(N)$. Despite its efficiency, previous works [4, 39, 15, 16] proved that linear attention suffers from insufficient expressive power, making it impractical for real applications. Surprisingly, we find a very close relationship between the formulas of high-performance Mamba

---

*Corresponding Author.

38th Conference on Neural Information Processing Systems (NeurIPS 2024).

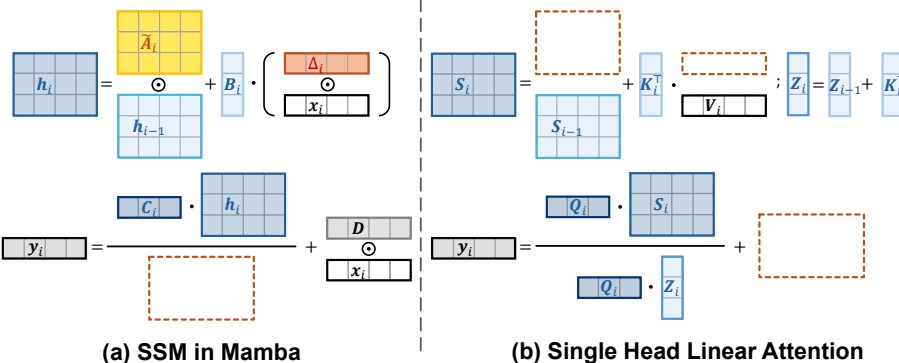

**(a) SSM in Mamba**          **(b) Single Head Linear Attention**

Figure 1: Illustration of selective SSM in Mamba (eq. (11)) and single head linear attention (eq. (12)). It can be seen that selective SSM resembles single-head linear attention with additional input gate $\mathbf{\Delta}_i$, forget gate $\widetilde{\mathbf{A}}_i$ and shortcut $\mathbf{D} \odot \mathbf{x}_i$, while omitting normalization $\mathbf{Q}_i \mathbf{Z}_i$.

and subpar linear attention Transformer. Therefore, a compelling research question emerges: *What factors contribute to Mamba's success and its significant superiority to linear attention Transformer?*

In this paper, we offer both theoretical and empirical analyses to unveil Mamba through the lens of linear attention Transformer. Specifically, we rewrite the formulas of selective state space model and linear attention within a unified formulation, depicting Mamba as a variation of linear attention Transformer with six distinctions: input gate, forget gate, shortcut, no attention normalization, single-head, and modified block design. To demystify what factors lead to Mamba's effectiveness, empirical studies on vision tasks are conducted to assess the impact of each special design. The results demonstrate that the forget gate and block design tend to be the two core contributors to Mamba's superiority. While the block design can be easily adopted, the forget gate necessitates recurrent computation, which may not be well-suited for non-auto-regressive vision models. Therefore, we delve into the essence of the forget gate and verify that it can be replaced by suitable positional encoding in vision tasks. Based on our findings, we introduce the two core contributors or their alternatives to linear attention Transformer, presenting our **Mamba-Inspired Linear Attention (MILA)** model. Experimental results demonstrate that MILA achieves superior results to various vision Mamba models in both image classification and high-resolution dense prediction tasks, validating that linear attention can surpass Mamba with the merits of two core designs.

Our main contributions and takeaways are as follows:

- We reveal Mamba's close relationship to linear attention Transformer: *Mamba and linear attention Transformer can be formulated within a unified framework, with Mamba exhibiting six distinct designs compared to the conventional linear attention paradigm: input gate, forget gate, shortcut, no attention normalization, single-head and modified block design.*

- We provide detailed analyses of each special design and empirically validate that the *forget gate and block design largely lead to Mamba's superiority.* Additionally, we demonstrate that the recurrent calculation of the forget gate might not be ideal for vision models. Instead, proper positional encoding can function as the forget gate in vision tasks, while preserving parallelizable computation and fast inference speed.

- We develop a series of linear attention vision Transformer models named MILA, which inherit the core merits of Mamba and tend to be more suitable for vision tasks than the original Mamba model.

## 2 Related Works

**Vision Transformer and attention.** Originating from natural language processing, Transformer and attention have been highly successful in vision tasks [32, 3, 36, 37], demonstrating superiority to the conventional CNN models[18, 7, 12]. However, the quadratic complexity of widely adopted Softmax attention [42] poses challenges in handling high-resolution images. Numerous works have attempted to reduce the computational cost by introducing local attention windows [32, 9, 20] or sparsity [43, 45, 56]. Linear attention [26], another approach, inherently offers linear complexity $\mathcal{O}(N)$ and is

capable of modeling long sequences. Despite its efficiency, previous works [4, 39, 47, 17] have shown that linear attention always fails to deliver satisfactory results, limiting its applicability.

**Mamba** [14] is a recently proposed state space model that achieves effective sequence modeling with linear complexity. Motivated by its potential for modeling high-resolution images, many researchers try to apply Mamba to vision tasks [14, 31, 27, 25, 38, 49, 48, 23]. For instance, VMamba [31] introduces a cross-scan module to enable 1D selective scanning in 2D image space. LocalMamba [25] utilizes local windows to enhance local modeling capability. EfficientVMamba [38] designs an atrous-based selective scan approach to enhance efficiency. In addition, MambaOut [51] analyzes whether Mamba is needed for vision, and explainability methods [1] have also been proposed.

In contrast to incorporating Mamba into vision, this paper reveals the surprising similarities between the formulas of inferior linear attention Transformer and powerful Mamba model. This interesting finding gives us the opportunity to demystify the key factors behind Mamba's success.

## 3 Preliminaries

This section revisits the formulations of attention and selective state space model. To facilitate comparison in Sec. 4, we employ identical notations for the dimensions of certain variables in both linear attention and selective state space model, and make some modifications to the formula formats.

### 3.1 Attention Mechanism

Let $x \in \mathbb{R}^{N \times C}$ denote a sequence of $N$ features with dimension $C$. Single head **Softmax attention** [42], also known as dot-product attention, can be written as:

$$Q = x\mathbf{W}_Q, K = x\mathbf{W}_K, V = x\mathbf{W}_V, \quad y_i = \sum_{j=1}^{N} \frac{\exp\left(Q_i K_j^\top / \sqrt{d}\right)}{\sum_{j=1}^{N} \exp\left(Q_i K_j^\top / \sqrt{d}\right)} V_j, \quad (1)$$

where $\mathbf{W}_Q, \mathbf{W}_K \in \mathbb{R}^{C \times d}, \mathbf{W}_V \in \mathbb{R}^{C \times C}$ denote projection matrices, $Q, K \in \mathbb{R}^{N \times d}, V \in \mathbb{R}^{N \times C}$ represent query/key/value matrices, and $Q_i, K_i \in \mathbb{R}^{1 \times d}, V_i \in \mathbb{R}^{1 \times C}$ are individual query/key/value tokens. Softmax attention computes the similarities between each query-key pair, leading to $\mathcal{O}(N^2)$ complexity. Therefore, it incurs unbearable computational cost in long-sequence modeling scenarios.

**Linear attention** [26], another attention paradigm, is proposed to effectively address this problem by reducing the computation complexity to $\mathcal{O}(N)$. Specifically, linear attention replaces the non-linear Softmax function with linear normalization, and adopts an additional kernel function $\phi$ in $Q$ and $K$:

$$Q = \phi(x\mathbf{W}_Q), K = \phi(x\mathbf{W}_K), V = x\mathbf{W}_V, \quad y_i = \sum_{j=1}^{N} \frac{Q_i K_j^\top}{\sum_{j=1}^{N} Q_i K_j^\top} V_j = \frac{Q_i\left(\sum_{j=1}^{N} K_j^\top V_j\right)}{Q_i\left(\sum_{j=1}^{N} K_j^\top\right)}. \quad (2)$$

This enables the rearrangement of the computation order from $(QK^\top)V$ to $Q(K^\top V)$ based on the associative property of matrix multiplication, thus reducing computation complexity to $\mathcal{O}(N)$.

Equation (2) defines linear attention with a global receptive field, where each query aggregates information from all keys and values. In practice, linear attention can also be implemented in autoregressive models, restricting the receptive field of the $i$-th token to proceeding tokens, i.e., token $j, j \leq i$. This causal linear attention is formulated as follows:

$$y_i = \frac{Q_i\left(\sum_{j=1}^{i} K_j^\top V_j\right)}{Q_i\left(\sum_{j=1}^{i} K_j^\top\right)} \triangleq \frac{Q_i S_i}{Q_i Z_i}, \quad S_i = \sum_{j=1}^{i} K_j^\top V_j, \quad Z_i = \sum_{j=1}^{i} K_j^\top. \quad (3)$$

This results in a recurrent linear attention form:

$$S_i = S_{i-1} + K_i^\top V_i, \quad Z_i = Z_{i-1} + K_i^\top, \quad y_i = Q_i S_i / Q_i Z_i. \quad (4)$$

## 3.2 Selective State Space Model

**State space model (SSM).** The classical state space model is a continuous system that maps the input $x(t) \in \mathbb{R}$ to output $y(t) \in \mathbb{R}$ through a hidden state $\boldsymbol{h}(t) \in \mathbb{R}^{d \times 1}$, which can be written as follows:

$$
\begin{aligned}
\boldsymbol{h}'(t) &= \boldsymbol{A}\boldsymbol{h}(t) + \boldsymbol{B}x(t), & x(t) \in \mathbb{R}, \ \ \boldsymbol{A} \in \mathbb{R}^{d \times d}, \ \ \boldsymbol{B}, \boldsymbol{h}(t), \boldsymbol{h}'(t) \in \mathbb{R}^{d \times 1}, \\
y(t) &= \boldsymbol{C}\boldsymbol{h}(t) + Dx(t), & y(t) \in \mathbb{R}, \ \ \boldsymbol{C} \in \mathbb{R}^{1 \times d}, \ \ D \in \mathbb{R}.
\end{aligned}
\tag{5}
$$

**Discrete SSM.** To be applied to deep neural networks, SSM is first transformed into its discrete version through zero-order hold discretization. Specifically, the continuous parameters $\boldsymbol{A}, \boldsymbol{B}$ are transformed into their discretized counterparts $\overline{\boldsymbol{A}}, \overline{\boldsymbol{B}}$ using a timescale parameter $\Delta \in \mathbb{R}$:

$$
\overline{\boldsymbol{A}} = \exp(\Delta \boldsymbol{A}), \quad \overline{\boldsymbol{B}} = (\Delta \boldsymbol{A})^{-1}(\exp(\Delta \boldsymbol{A}) - \boldsymbol{I}) \cdot \Delta \boldsymbol{B} \approx \Delta \boldsymbol{B}.
\tag{6}
$$

Therefore, discrete SSM rewrite eq. (5) as:

$$
\begin{aligned}
\boldsymbol{h}_i &= \overline{\boldsymbol{A}}\boldsymbol{h}_{i-1} + \overline{\boldsymbol{B}}x_i, & x_i \in \mathbb{R}, \ \ \overline{\boldsymbol{A}} \in \mathbb{R}^{d \times d}, \ \ \overline{\boldsymbol{B}}, \boldsymbol{h}_{i-1}, \boldsymbol{h}_i \in \mathbb{R}^{d \times 1}, \\
y_i &= \boldsymbol{C}\boldsymbol{h}_i + Dx_i, & y_i \in \mathbb{R}, \ \ \boldsymbol{C} \in \mathbb{R}^{1 \times d}, \ \ D \in \mathbb{R}.
\end{aligned}
\tag{7}
$$

**Selective State Space Model.** Mamba [14] improves SSM with selection, presenting the selective state space model. The parameters $\boldsymbol{B}, \boldsymbol{C}, \Delta$ is set as the function of $x_i$, thus becoming input-dependent parameters $\boldsymbol{B}_i, \boldsymbol{C}_i, \Delta_i$. As a result, the discretized parameters $\overline{\boldsymbol{A}}_i = \exp(\Delta_i \boldsymbol{A})$, $\overline{\boldsymbol{B}}_i = \Delta_i \boldsymbol{B}_i$ are also input-dependent. The selective state space model can be written as:

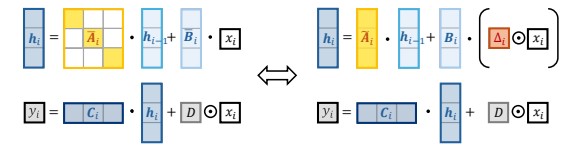

**(a) Selective SSM Model**  **(b) Equivalent Form**

Figure 2: Illustration of selective state space model (eq. (8)) and its equivalent form (eq. (9)).

$$
\begin{aligned}
\boldsymbol{h}_i &= \overline{\boldsymbol{A}}_i \boldsymbol{h}_{i-1} + \overline{\boldsymbol{B}}_i x_i, & x_i \in \mathbb{R}, \ \ \overline{\boldsymbol{A}}_i \in \mathbb{R}^{d \times d}, \ \ \overline{\boldsymbol{B}}_i, \boldsymbol{h}_{i-1}, \boldsymbol{h}_i \in \mathbb{R}^{d \times 1}, \\
y_i &= \boldsymbol{C}_i \boldsymbol{h}_i + Dx_i, & y_i \in \mathbb{R}, \ \ \boldsymbol{C}_i \in \mathbb{R}^{1 \times d}, \ \ D \in \mathbb{R}.
\end{aligned}
\tag{8}
$$

For the convenience of subsequent derivation, we make three modifications to eq. (8):

- Mamba practically sets $\boldsymbol{A}, \overline{\boldsymbol{A}}_i$ as diagonal matrices. Therefore, $\overline{\boldsymbol{A}}_i \boldsymbol{h}_{i-1} = \widetilde{\boldsymbol{A}}_i \odot \boldsymbol{h}_{i-1}$, where $\widetilde{\boldsymbol{A}}_i = \mathrm{diag}(\overline{\boldsymbol{A}}_i) \in \mathbb{R}^{d \times 1}$ denotes the matrix composed of diagonal elements of $\overline{\boldsymbol{A}}_i$.
- Given $\overline{\boldsymbol{B}}_i = \Delta_i \boldsymbol{B}_i$ and $\Delta_i \in \mathbb{R}$, we have $\overline{\boldsymbol{B}}_i x_i = \Delta_i \boldsymbol{B}_i x_i = \boldsymbol{B}_i(\Delta_i x_i) = \boldsymbol{B}_i(\Delta_i \odot x_i)$.
- $Dx_i = D \odot x_i$, where $\odot$ denotes the Hadamard product, i.e., element-wise multiplication.

Consequently, we rewrite eq. (8) as:

$$
\begin{aligned}
\boldsymbol{h}_i &= \widetilde{\boldsymbol{A}}_i \odot \boldsymbol{h}_{i-1} + \boldsymbol{B}_i(\Delta_i \odot x_i), & x_i, \Delta_i \in \mathbb{R}, \ \ \widetilde{\boldsymbol{A}}_i, \boldsymbol{B}_i, \boldsymbol{h}_{i-1}, \boldsymbol{h}_i \in \mathbb{R}^{d \times 1}, \\
y_i &= \boldsymbol{C}_i \boldsymbol{h}_i + D \odot x_i, & y_i \in \mathbb{R}, \ \ \boldsymbol{C}_i \in \mathbb{R}^{1 \times d}, \ \ D \in \mathbb{R}.
\end{aligned}
\tag{9}
$$

The selective state space model formulated in eq. (9) can only deal with scalar input $x_i \in \mathbb{R}$. To operate over an input sequence $\boldsymbol{x} \in \mathbb{R}^{N \times C}, \boldsymbol{x}_i \in \mathbb{R}^{1 \times C}$, Mamba applies eq. (9) independently to each channel, leading to the following formulations:

$$
\begin{aligned}
\boldsymbol{h}_i &= \widetilde{\boldsymbol{A}}_i \odot \boldsymbol{h}_{i-1} + \boldsymbol{B}_i(\boldsymbol{\Delta}_i \odot \boldsymbol{x}_i), & \boldsymbol{x}_i, \boldsymbol{\Delta}_i \in \mathbb{R}^{1 \times C}, \ \ \widetilde{\boldsymbol{A}}_i, \boldsymbol{h}_{i-1}, \boldsymbol{h}_i \in \mathbb{R}^{d \times C}, \ \ \boldsymbol{B}_i \in \mathbb{R}^{d \times 1} \\
\boldsymbol{y}_i &= \boldsymbol{C}_i \boldsymbol{h}_i + \boldsymbol{D} \odot \boldsymbol{x}_i, & \boldsymbol{y}_i \in \mathbb{R}^{1 \times C}, \ \ \boldsymbol{C}_i \in \mathbb{R}^{1 \times d}, \ \ \boldsymbol{D} \in \mathbb{R}^{1 \times C},
\end{aligned}
\tag{10}
$$

where $\boldsymbol{B}_i, \boldsymbol{C}_i, \boldsymbol{\Delta}_i$ are derived from the input. Specifically, Mamba employs $\boldsymbol{B} = (\boldsymbol{x}\mathbf{W}_B)^{\top}$, $\boldsymbol{C} = \boldsymbol{x}\mathbf{W}_C$, $\boldsymbol{\Delta} = \mathrm{Softplus}(\boldsymbol{x}\mathbf{W}_1\mathbf{W}_2)$ to produce the parameters $\boldsymbol{B} \in \mathbb{R}^{d \times N}, \boldsymbol{C} \in \mathbb{R}^{N \times d}, \boldsymbol{\Delta} \in \mathbb{R}^{N \times C}$, where $\mathbf{W}_B, \mathbf{W}_C \in \mathbb{R}^{C \times d}, \mathbf{W}_1 \in \mathbb{R}^{C \times C_0}, \mathbf{W}_2 \in \mathbb{R}^{C_0 \times C}$ are projection matrices. *Notably, eq. (10) is exactly the selective SSM employed in Mamba, we only make modifications to formula formats.*

## 4 Connecting Mamba and Linear Attention Transformer

In this section, we reveal the similarities and disparities between Mamba and linear attention Transformer from two perspectives: core operation and macro architecture.

## 4.1 Interpreting Selective State Space Model as Linear Attention

As detailed in Sec. 3, for an input sequence of $N$ tokens $\boldsymbol{x} \in \mathbb{R}^{N \times C}$, the formulations of selective state space model and linear attention are provided by eq. (10) and eq. (4), respectively. Many underlying similarities exist between the formulas of these two operations. To facilitate comprehension, we rewrite eq. (10) and eq. (4) with a unified formulation as follows:

$$
\begin{aligned}
\boldsymbol{h}_i &= \widetilde{\boldsymbol{A}}_i \odot \boldsymbol{h}_{i-1} + \boldsymbol{B}_i(\boldsymbol{\Delta}_i \odot \boldsymbol{x}_i), \\
\boldsymbol{y}_i &= \boldsymbol{C}_i \boldsymbol{h}_i \ / \ 1 + \boldsymbol{D} \odot \boldsymbol{x}_i.
\end{aligned} \quad (11) \qquad
\begin{aligned}
\boldsymbol{S}_i &= \boldsymbol{1} \odot \boldsymbol{S}_{i-1} + \boldsymbol{K}_i^\top(\boldsymbol{1} \odot \boldsymbol{V}_i), \\
\boldsymbol{y}_i &= \boldsymbol{Q}_i \boldsymbol{S}_i \ / \ \boldsymbol{Q}_i \boldsymbol{Z}_i + \boldsymbol{0} \odot \boldsymbol{x}_i.
\end{aligned} \quad (12)
$$

As illustrated in Fig. 1, a close relationship between eq. (11) and eq. (12) is evident. Specifically, $\boldsymbol{h}_i \sim \boldsymbol{S}_i \in \mathbb{R}^{d \times C}$, $\boldsymbol{B}_i \sim \boldsymbol{K}_i^\top \in \mathbb{R}^{d \times 1}$, $\boldsymbol{x}_i \sim \boldsymbol{V}_i \in \mathbb{R}^{1 \times C}$, and $\boldsymbol{C}_i \sim \boldsymbol{Q}_i \in \mathbb{R}^{1 \times d}$. Therefore, selective SSM can be viewed as a special variation of linear attention, indicating a very close connection between these two mechanisms. Furthermore, four major differences can be observed:

1. In eq. (11), the input $\boldsymbol{x}_i$ is augmented by Hadamard product with $\boldsymbol{\Delta}_i$. Since $\boldsymbol{\Delta} = \mathrm{Softplus}(\boldsymbol{x}\mathbf{W}_1\mathbf{W}_2)$, all elements of $\boldsymbol{\Delta}_i$ are positive. Therefore, we view $\boldsymbol{\Delta}_i$ as an *input gate*, controlling whether to let the input $\boldsymbol{x}_i$ into the hidden state.

2. There is an additional $\widetilde{\boldsymbol{A}}_i$ in eq. (11). Mamba sets $\boldsymbol{A}$ as a diagonal matrix with negative diagonal elements, thus ensuring all elements of $\widetilde{\boldsymbol{A}}_i = \mathrm{diag}(\overline{\boldsymbol{A}}_i) = \exp(\mathrm{diag}(\boldsymbol{A})\boldsymbol{\Delta}_i)$ to fall between 0 and 1. Hence, we interpret $\widetilde{\boldsymbol{A}}_i$ as a *forget gate*, which decides the degree of attenuation for the previous hidden state $\boldsymbol{h}_{i-1}$.

3. A learnable *shortcut* from the input $\boldsymbol{x}_i$ to the output $\boldsymbol{y}_i$ is employed in eq. (11), i.e. $\boldsymbol{D} \odot \boldsymbol{x}_i$.

4. As depicted in eq. (12), linear attention divides the output by $\boldsymbol{Q}_i \boldsymbol{Z}_i$ to maintain that the attention weights sum up to 1, while eq. (11) does not have such *normalization*.

In addition to these four differences, it is also important to note that eq. (12) represents *single-head* linear attention as there is only one group of $Q, K$. This indicates that the selective state space model is akin to *single-head* linear attention and does not incorporate a *multi-head* design.

In a word, the similarities and disparities between selective SSM and linear attention can be summarized as: ***selective state space model resembles linear attention with additional input gate, forget gate and shortcut, while omitting normalization and multi-head design.***

## 4.2 Analysis of Differences in Core Operations

**Input gate.** As discussed before, $\boldsymbol{\Delta}_i$ actually functions as an input gate for $\boldsymbol{x}_i$, determining its access to the hidden state. The values of this input gate are predicted from the current input $\boldsymbol{x}_i$ as $\boldsymbol{\Delta}_i = \mathrm{Softplus}(\boldsymbol{x}_i\mathbf{W}_1\mathbf{W}_2)$. Therefore, by learning the weight of $\mathbf{W}_1, \mathbf{W}_2$, the model can discern the "utility" of $\boldsymbol{x}_i$, generating large $\boldsymbol{\Delta}_i$ values for "useful" $\boldsymbol{x}_i$ and small ones for "less useful" $\boldsymbol{x}_i$. For example, in vision tasks, tokens representing foreground objects may yield larger input gate values, while background tokens may yield smaller ones.

**Forget gate.** $\widetilde{\boldsymbol{A}}_i$ acts as a forget gate in selective state space model, offering two essential properties: local bias and positional information. Firstly, all elements of $\widetilde{\boldsymbol{A}}_i$ strictly range from 0 to 1, indicating that the model consistently decays the previous hidden state $\boldsymbol{h}_{i-1}$ upon the arrival of the current token $\boldsymbol{x}_i$. This results in a strong local bias. Secondly, $\widetilde{\boldsymbol{A}}_i$ provides positional information for the model. It ensures that the model is sensitive to the order of input sequences. Without this forget gate, rearranging the order of the preceding sequence will not affect subsequent outputs. For instance, in recurrent linear attention, if we change the order of $\boldsymbol{x}_1$ and $\boldsymbol{x}_2$, the outputs $\boldsymbol{y}_i, i \geq 3$ will not change. Hence, the forget gate $\widetilde{\boldsymbol{A}}_i$ plays an important role in selective SSM.

Despite its effectiveness, incorporating the forget gate also poses significant challenges. Firstly, it forces the model to adopt the recurrent formulation during *both training and inference*. Previous state space models typically use global convolution for efficient parallelizable training, which is incompatible with selective SSM due to the input-dependency of $\widetilde{\boldsymbol{A}}_i$. As a remedy, Mamba [14] proposes a hardware-aware algorithm to speed up computation by performing parallel scan in *recurrent mode* (see the abstract of [14]). Although effective, such recurrent calculation unavoidably reduces model throughput and is still slower than parallel linear attention (eq. (2)). Secondly, the

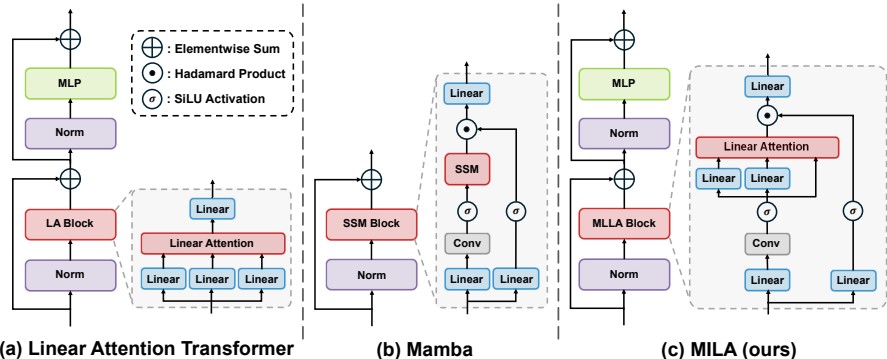

Figure 3: Illustration of the macro designs of linear attention Transformer, Mamba and our MILA.

forget gate inherently functions in causal mode, which may not be very suitable for non-auto-regressive vision models. Using the forget gate $\widetilde{A}_i$ in vision tasks requires transforming the image into a 1D sequence and conducting recurrent computation, which limits the receptive field of each image token to its preceding sequence and incurs extra latency. Therefore, we believe that the forget gate is ideally suited for modeling causal data, which naturally needs auto-regressive training and recurrent calculation. However, it may not be as suitable for non-causal data like images. We further speculate that a suitable positional encoding can substitute for the forget gate, since certain positional encodings, such as LePE [9] and RoPE [41], can also provide local bias and positional information.

**Shortcut.** Selective SSM employs a learnable shortcut $D \odot x_i$, making it resemble a residual block [22]. This shortcut may aid in optimizing the model and stabilizing training.

**Normalization.** The output in linear attention is divided by $Q_i Z_i$ to ensure the attention weights sum up to 1. We believe this normalization is crucial for stabilizing training and improving model capacity. Let's consider an input $\alpha x_i, \alpha > 0$. It is transformed into $\alpha Q_i, \alpha K_i, \alpha V_i$ through projections. If there is no normalization on attention weights, as $\alpha$ increases, $\alpha Q_i$ exhibits larger similarities with all keys $\alpha Q_i K_j^\top, \forall j$. This indicates that longer tokens will have larger attention scores with every token, leading to longer output. Additionally, as $\alpha$ grows bigger, $\alpha K_i$ yields bigger similarities with all queries $\alpha Q_j K_i^\top, \forall j$. This implies that all queries will focus more on longer tokens. As a result, longer tokens tend to dominate the whole feature map, while shorter tokens may fail to represent their corresponding semantics. This may result in training instability and could possibly lower model's expressiveness. Normalizing the attention weights can significantly alleviate this issue.

**Multi-head.** Linear attention commonly utilizes multi-head design [42] for better outcome. Multi-head attention employs multiple groups of $Q, K$ to produce attention matrices and allows the model to simultaneously attend to information from various representation subspaces at different positions, thus enhancing its expressive power.

### 4.3 Analysis of Macro Architecture Design

Modern linear attention Transformer models commonly adopt the block design depicted in Fig. 3(a), which are comprised of a linear attention sub-block and a MLP sub-block. In contrast, Mamba modifies the block design by combining two basic designs, H3 [13] and Gated Attention [24], resulting in the architecture illustrated in Fig. 3(b). The improved Mamba block integrates multiple operations such as selective SSM, depth-wise convolution, linear mapping, activation function, gating mechanism, etc., and tends to be more effective than the conventional Transformer block design.

### 4.4 Relationship between Mamba and Linear Attention Transformer

***Mamba can be seen as a variant of linear attention Transformer with specialized linear attention and modified block design.*** The special linear attention variation, i.e. selective state space model, has five major distinctions from the common linear attention paradigm, detailed in Sec. 4.2. And the differences in block designs are analyzed in Sec. 4.3. In summary, Sec. 4.2 and Sec. 4.3 reveal the intimate relationship between Mamba and linear attention Transformer, highlighting a total of six differences: five in core operation and one in macro design.

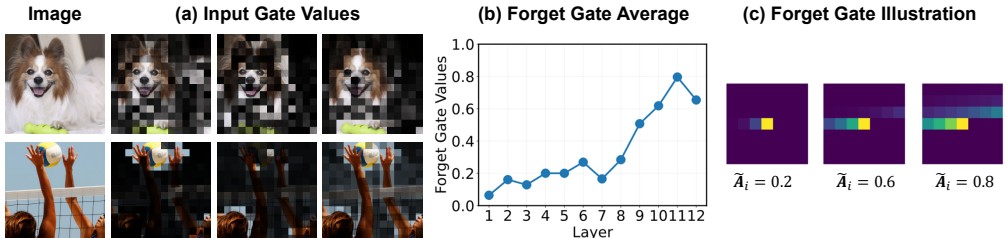

Figure 4: (a) Visualizations of the distributions of input gate values. (b) The average of forget gate values in different layers. (c) The attenuation effect of different forget gate values.

## 5 Empirical Study

Mamba [14] is seen as a powerful alternative to Transformer [42], while linear attention models generally being considered inferior [39, 15]. In Sec. 4, we illustrated the surprisingly close relationship between Mamba and linear attention Transformer and pointed out six major distinctions. In this section, we conduct experiments to assess the impact of each distinction, shedding some light on the core contributors behind Mamba's success.

### 5.1 Implementation

We employ the widely used Swin Transformer [32] architecture to verify the effects of the six differences. Firstly, we substitute the Softmax attention in Swin Transformer with linear attention to create our baseline model. Subsequently, we *separately* introduce each distinction to the baseline model to assess its impact. Based on the results, we further validate whether linear attention can achieve superior results with the merits of Mamba. Specifically, we integrate the useful designs into linear attention Transformer to create our **Mamba-Inspired Linear Attention (MILA)** model, and assess its effectiveness by comparing it with various vision Mamba designs across multiple tasks, including ImageNet-1K classification [8], COCO object detection [30], and ADE20K semantic segmentation [55]. Detailed model architectures and training setups are shown in the Appendix.

### 5.2 Empirical Analysis of the Differences

As shown in Tab. 1, we *separately* apply each distinction to the baseline linear attention model and assess their performances on ImageNet-1K.

**Input Gate.** Introducing the input gate results in a modest accuracy improvement of 0.2, indicating that it is slightly helpful for the model. Visualizations in Fig. 4(a) aid in understanding the impact of the input gate. It can be seen that the model tends to generate higher input gate values for more informative regions like fore-

Table 1: Ablation on the impact of each distinction.

| Architecture | #Params | FLOPs | Throughput | Top-1 |
|---|---|---|---|---|
| Baseline | 28M | 4.5G | 1152 | 77.6 |
| + Input Gate | 29M | 4.5G | 1069 | 77.8 |
| + Forget Gate | 29M | 4.8G | 743 | 78.4 |
| + Shortcut | 28M | 4.5G | 1066 | 77.8 |
| − Normalization | 28M | 4.5G | 1215 | 72.4 |
| − Multi-head Design | 24M | 3.9G | 1540 | 73.5 |
| + Block Design all. | 28M | 4.7G | 915 | 79.4 |
| + Block Design sub. | 31M | 4.8G | 1010 | 80.9 |

ground objects, while suppressing less useful tokens. However, the model struggles to generate highly effective input gates, since the input gate values $\mathbf{\Delta}_i = \mathrm{Softplus}(\boldsymbol{x}_i \mathbf{W}_1 \mathbf{W}_2)$ are predicted solely from the current input token $\boldsymbol{x}_i$ without considering the overall semantics of the image. For example, in one image, the dog may be the area of interest, whereas in another, it might simply be part of the background. Without leveraging information from the entire image, assigning large input gate values to the dog in one image while blocking it in another is impractical. Moreover, employing input gate results in a 7% decrease in model throughput.

**Forget Gate.** Employing the forget gate in linear attention leads to an obvious performance improvement from 77.6 to 78.4. However, such accuracy gain comes at a cost: the model throughput drops severely from 1152 to 743. This is because the forget gate has to employ recurrent calculation, which is slower than the parallelizable matrix multiplication in linear attention. It's worth noting that we already utilize the hardware-aware algorithm proposed in Mamba to speed up the recurrent computation. Thus, we believe the forget gate might not be very suitable for modeling non-causal data like

Table 3: Comparison with SOTA Vision Mambas on ImageNet-1K.

| Method | Type | #Params | FLOPs | Top-1 | Method | Type | #Params | FLOPs | Top-1 |
|---|---|---|---|---|---|---|---|---|---|
| ConvNeXt-T [33] | CNN | 29M | 4.5G | 82.1 | ConvNeXt-S [33] | CNN | 50M | 8.7G | 83.1 |
| MambaOut-T [51] | CNN | 27M | 4.5G | 82.7 | MambaOut-S [51] | CNN | 48M | 9.0G | 84.1 |
| Swin-T [32] | Transformer | 29M | 4.5G | 81.3 | PVTv2-B3 [44] | Transformer | 45M | 7.9G | 83.2 |
| PVTv2-B2 [44] | Transformer | 25M | 4.0G | 82.0 | CSwin-S [9] | Transformer | 35M | 6.9G | 83.6 |
| Focal-T [50] | Transformer | 29M | 4.9G | 82.2 | Focal-S [50] | Transformer | 51M | 9.4G | 83.6 |
| MViTv2-T [28] | Transformer | 24M | 4.7G | 82.3 | MViTv2-S [28] | Transformer | 35M | 7.0G | 83.6 |
| CSwin-T [9] | Transformer | 23M | 4.3G | 82.7 | VMamba-S [31] | Mamba | 50M | 8.7G | 83.6 |
| DiNAT-T [19] | Transformer | 28M | 4.3G | 82.7 | LocalVMamba-S [25] | Mamba | 50M | 11.4G | 83.7 |
| NAT-T [20] | Transformer | 28M | 4.3G | 83.2 | MILA-S | MILA | 43M | 7.3G | 84.4 |
| PlainMamba-L1 [49] | Mamba | 7M | 3.0G | 77.9 | ConvNeXt-B [33] | CNN | 89M | 15.4G | 83.8 |
| Vim-S [57] | Mamba | 26M | 5.1G | 80.3 | MambaOut-B [51] | CNN | 85M | 15.8G | 84.2 |
| LocalVim-S [25] | Mamba | 28M | 4.8G | 81.2 | PVTv2-B5 [44] | Transformer | 82M | 11.8G | 83.8 |
| PlainMamba-L2 [49] | Mamba | 25M | 8.1G | 81.6 | Focal-B [50] | Transformer | 90M | 16.4G | 84.0 |
| Mamba2D-S [27] | Mamba | 24M | – | 81.7 | CSwin-B | Transformer | 78M | 15.0G | 84.2 |
| EfficientVMamba-B [38] | Mamba | 33M | 4.0G | 81.8 | NAT-B [20] | Transformer | 90M | 13.7G | 84.3 |
| VMamba-T [31] | Mamba | 31M | 4.9G | 82.5 | PlainMamba-L3 [49] | Mamba | 50M | 14.4G | 82.3 |
| LocalVMamba-T [25] | Mamba | 26M | 5.7G | 82.7 | Mamba2D-B [27] | Mamba | 94M | – | 83.0 |
|  |  |  |  |  | VMamba-B [31] | Mamba | 89M | 15.4G | 83.9 |
| MILA-T | MILA | 25M | 4.2G | 83.5 | MILA-B | MILA | 96M | 16.2G | 85.3 |

images, which do not inherently require recurrence. As an alternative, we analyze the fundamental properties of the forget gate and attempt to substitute it with other parallelizable operations.

In Fig. 4, we calculate the average of forget gate values in each layer and illustrate the attenuation effect of different forget gate values. In shallow layers, the forget gate values $\widetilde{A}_i \approx 0.2$, indicating that each token primarily focuses on itself and the preceding two tokens, demonstrating strong local bias. In deeper layers, the average is approximately 0.6-0.8, suggesting a broad receptive field for each token. This confirms our previous analysis that the forget gate offers two crucial properties for the model, namely local bias and positional information. We conduct

Table 2: Substituting the forget gate with various positional encodings.

|  | #Params | FLOPs | Throughput | Top-1 |
|---|---|---|---|---|
| Baseline | 28M | 4.5G | 1152 | 77.6 |
| + Forget Gate | 29M | 4.8G | 743 | 78.4 |
| + APE [10] | 30M | 4.5G | 1132 | 80.0 |
| + LePE [9] | 28M | 4.5G | 1074 | 81.6 |
| + CPE [5] | 28M | 4.5G | 1099 | 81.7 |
| + RoPE [41] | 28M | 4.5G | 1113 | 80.0 |

experiments to verify whether the forget gate can be replaced with proper positional encoding, which can also provide local bias and positional information. Results in Tab. 2 show that APE [42], LePE [9], CPE [5] and RoPE [41] can both help the model yield better results than the forget gate, while maintaining high throughput. We attribute the improved outcomes to a broader receptive field. Specifically, when using the forget gate, we have to adopt the recurrent linear attention format which restricts the receptive field of each token to the preceding sequence. In contrast, without the forget gate, it is natural to utilize parallel linear attention to achieve a global receptive field.

**Shortcut.** As illustrated in Tab. 1, the usage of learnable shortcut in linear attention provides a 0.2 accuracy gain, while decreasing the throughput from 1152 to 1066.

**Normalization.** Without normalization, the model suffers from severe performance degradation from 77.6 to 72.4. This can be attributed to the issue of long tokens dominating, as discussed in Sec. 4.2. To confirm this, we compute the standard deviation of token lengths (l2 norm) in each layer using both the baseline model and the model without attention normalization. As depicted in Fig. 5, without normalization, the standard deviation of token lengths tends to be much larger than the baseline, particularly in the last two layers. This supports our analysis that without normalization, the difference in token length becomes significant, with some long tokens dominating the model while others struggling to convey their semantics.

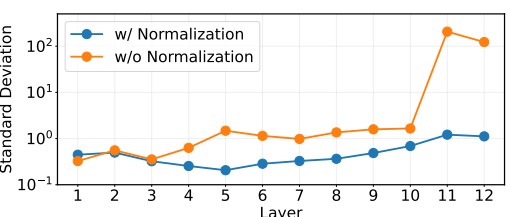

Figure 5: The standard deviation of token lengths.

**Multi-head.** Modern Transformers typically adopt the multi-head design [42] to enhance their expressive power. As shown in Tab. 1, removing this design reduces computational cost and accelerates the model but significantly diminishes performance. We consider this trade-off unwarranted.

**Block Design.** We employ two ways to assess the effects of Mamba's block design: 1. Substituting the entire Transformer block with Mamba's block architecture. 2. Replacing the attention sub-block with Mamba's block design, while preserving the MLP sub-block. In both settings, the selective SMM in Mamba's block is substituted with linear attention. To maintain similar FLOPs, we employ Mamba expansion factors [14] $E = 2.0$ and $E = 1.0$ for the two settings, respectively. The results are presented in Tab. 1 as "Block Design all" and "Block Design sub". Both replacement approaches result in performance improvements, demonstrating the efficacy of Mamba's macro design. Substituting the attention sub-block yields better result, which creates our MILA block shown in Fig. 3(c). Notably, we omit the $V$ projection before linear attention calculation, as a similar input projection already exists. The module complexity of a MILA block is expressed as:

$$\Omega(\text{MILA}) = \underbrace{2NC^2 + 2NC^2 + NC^2}_{\text{In/Out, Q/K, Gate Projection}} + \underbrace{2NCd}_{\text{Linear Attention}} + \underbrace{k^2NC}_{\text{DWConv}} + \underbrace{8NC^2}_{\text{MLP}}, \quad (13)$$

which is slightly larger than the complexity of a Transformer block (Fig. 3a), $4NC^2 + 2NCd + 8NC^2$.

## 5.3 Comparison with Mamba in Vision

Based on our findings, we integrate the forget gate and block design into linear attention, introducing our MILA model. Notably, we practically use LePE, CPE, and RoPE to replace the forget gate's local bias, input-dependent positional information, and global positional information, respectively.

**ImageNet classification.** As shown in Tab. 3, our MILA models consistently outperform various vision Mamba models across all model sizes, owing to the integration of useful designs from both Mamba and linear attention. These results also validate that with the merits of Mamba's two key designs, the inferior linear attention Transformer can surpass high-performance Mamba. Notably, we empirically observe that MILA exhibits greater scalability compared to vision Mamba models, as MILA-B achieves an accuracy of 85.3, surpassing other models by a significant margin. Additionally, MILA also outperforms various CNN and vision Transformer designs. For instance, MILA exhibits better performance than MambaOut [51], a recent work that removes the selective SSM in Mamba and employs a gated convolution architecture.

**Inference time.** We offer real speed measurements in Fig. 6. Substituting the forget gate with positional encoding, our MILA models benefit from parallelizable computation, resulting in significantly faster inference speeds compared to vision Mamba models. For instance, our model achieves 4.5x faster inference speed than Mamba2D [27], while maintaining better accuracy. Compared to the highly optimized VMamba model [31], our model also delivers a 1.5x speedup accompanied by a 0.5 accuracy gain. These sub-

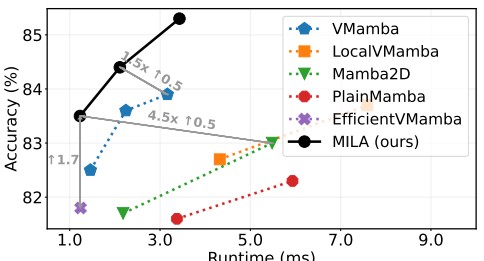

Figure 6: Speed tests on a RTX3090 GPU.

stantial improvements in model speed further support our analysis that the parallelizable MILA is more suitable than Mamba for modeling non-causal data such as images.

**COCO object detection.** As shown in Tab. 4, on the COCO dataset, MILA models also achieve superior results to vision Mamba models, implying their effectiveness in high-resolution dense prediction tasks. MILA offers effective *global modeling* with *linear complexity* $\mathcal{O}(N)$ (see eq. (13)) and *parallelizable computation*, making it ideally suitable for high-resolution image modeling scenarios. Notably, MILA outperforms MambaOut [51] by a significant margin, which aligns with the findings in MambaOut [51].

Table 5: Results of semantic segmentation using UperNet [46]. The FLOPs are computed with input resolution of $512 \times 2048$.

| Semantic Segmentation on ADE20K | | | | |
|---|---|---|---|---|
| Backbone | #Params | FLOPs | mIoU SS | MS |
| Swin-B [32] | 121M | 1188G | 48.1 | 49.7 |
| MambaOut-B [51] | 112M | 1178G | 49.6 | 51.0 |
| VMamba-B [31] | 122M | 1170G | 51.0 | 51.6 |
| MILA-B | 128M | 1183G | **51.9** | **52.5** |

**ADE-20K semantic segmentation.** We report the results on ADE-20K dataset in Tab. 5, where "SS" and "MS" denote single-scale and multi-scale testing, respectively. Similar to the object detection task, MILA yields better results in semantic segmentation, further verifying our analyses and the effectiveness of MILA model.

Table 4: Results on COCO dataset. The FLOPs are computed over backbone, FPN and detection head with an input resolution of 1280×800.

**(a) Mask R-CNN 1x on COCO**

| Method | Type | #Params | FLOPs | $AP^b$ | $AP^b_{50}$ | $AP^b_{75}$ | $AP^m$ | $AP^m_{50}$ | $AP^m_{75}$ |
|---|---|---|---|---|---|---|---|---|---|
| ConvNeXt-T [33] | CNN | 48M | 262G | 44.2 | 66.6 | 48.3 | 40.1 | 63.3 | 42.8 |
| MambaOut-T [51] | CNN | 43M | 262G | 45.1 | 67.3 | 49.6 | 41.0 | 64.1 | 44.1 |
| Swin-T [32] | Transformer | 48M | 267G | 42.7 | 65.2 | 46.8 | 39.3 | 62.2 | 42.2 |
| PVTv2-B2 [44] | Transformer | 45M | 309G | 45.3 | 67.1 | 49.6 | 41.2 | 64.2 | 44.4 |
| FocalNet-T [50] | Transformer | 49M | 268G | 46.1 | 68.2 | 50.6 | 41.5 | 65.1 | 44.5 |
| CSWin-T [9] | Transformer | 42M | 279G | 46.7 | 68.6 | 51.3 | 42.2 | 65.6 | 45.4 |
| EfficientVMamba-B [38] | Mamba | 53M | 252G | 43.7 | 66.2 | 47.9 | 40.2 | 63.3 | 42.9 |
| PlainMamba-Adapter-L1 [49] | Mamba | 31M | 388G | 44.1 | 64.8 | 47.9 | 39.1 | 61.6 | 41.9 |
| LocalVMamba-T [25] | Mamba | 45M | 291G | 46.7 | 68.7 | 50.8 | 42.2 | 65.7 | 45.5 |
| MILA-T | MILA | 44M | 255G | 46.8 | 69.5 | 51.5 | 42.1 | 66.4 | 45.0 |
| ConvNeXt-S [33] | CNN | 70M | 348G | 45.4 | 67.9 | 50.0 | 41.8 | 65.2 | 45.1 |
| MambaOut-S [51] | CNN | 65M | 354G | 47.4 | 69.1 | 52.4 | 42.7 | 66.1 | 46.2 |
| Swin-S [32] | Transformer | 69M | 354G | 44.8 | 66.6 | 48.9 | 40.9 | 63.2 | 44.2 |
| PVTv2-B3 [44] | Transformer | 65M | 397G | 47.0 | 68.1 | 51.7 | 42.5 | 65.7 | 45.7 |
| FocalNet-S [50] | Transformer | 72M | 365G | 48.3 | 70.5 | 53.1 | 43.1 | 67.4 | 46.2 |
| CSWin-S [9] | Transformer | 54M | 342G | 47.9 | 70.1 | 52.6 | 43.2 | 67.1 | 46.2 |
| PlainMamba-Adapter-L2 [49] | Mamba | 53M | 542G | 46.0 | 66.9 | 50.1 | 40.6 | 63.8 | 43.6 |
| LocalVMamba-S [25] | Mamba | 69M | 414G | 48.4 | 69.9 | 52.7 | 43.2 | 66.7 | 46.5 |
| Vmamba-S [31] | Mamba | 64M | 357G | 48.7 | 70.0 | 53.4 | 43.7 | 67.3 | 47.0 |
| MILA-S | MILA | 63M | 319G | 49.2 | 71.5 | 53.9 | 44.2 | 68.5 | 47.2 |
| ConvNeXt-B [33] | CNN | 108M | 486G | 47.0 | 69.4 | 51.7 | 42.7 | 66.3 | 46.0 |
| MambaOut-B [51] | CNN | 100M | 495G | 47.4 | 69.3 | 52.2 | 43.0 | 66.4 | 46.3 |
| Swin-B [32] | Transformer | 107M | 496G | 46.9 | – | – | 42.3 | – | – |
| PVTv2-B5 [44] | Transformer | 102M | 557G | 47.4 | 68.6 | 51.9 | 42.5 | 65.7 | 46.0 |
| FocalNet-B [50] | Transformer | 111M | 507G | 49.0 | 70.9 | 53.9 | 43.5 | 67.9 | 46.7 |
| CSWin-B [9] | Transformer | 97M | 526G | 48.7 | 70.4 | 53.9 | 43.9 | 67.8 | 47.3 |
| PlainMamba-Adapter-L3 [49] | Mamba | 79M | 696G | 46.8 | 68.0 | 51.1 | 41.2 | 64.7 | 43.9 |
| VMamba-B [31] | Mamba | 108M | 485G | 49.2 | 70.9 | 53.9 | 43.9 | 67.7 | 47.6 |
| MILA-B | MILA | 115M | 502G | 50.5 | 72.0 | 55.4 | 45.0 | 69.3 | 48.6 |

**(b) Mask R-CNN 3x on COCO**

| Method | Type | #Params | FLOPs | $AP^b$ | $AP^b_{50}$ | $AP^b_{75}$ | $AP^m$ | $AP^m_{50}$ | $AP^m_{75}$ |
|---|---|---|---|---|---|---|---|---|---|
| ConvNeXt-T [33] | CNN | 48M | 262G | 46.2 | 67.9 | 50.8 | 41.7 | 65.0 | 44.9 |
| Swin-T [32] | Transformer | 48M | 267G | 46.0 | 68.1 | 50.3 | 41.6 | 65.1 | 44.9 |
| PVTv2-B2 [44] | Transformer | 45M | 309G | 47.8 | 69.7 | 52.6 | 43.1 | 66.8 | 46.7 |
| FocalNet-T [50] | Transformer | 49M | 268G | 48.0 | 69.7 | 53.0 | 42.9 | 66.5 | 46.1 |
| Vmamba-T [31] | Mamba | 50M | 270G | 48.9 | 70.6 | 53.6 | 43.7 | 67.7 | 46.8 |
| LocalVMamba-T [25] | Mamba | 45M | 291G | 48.7 | 70.1 | 53.0 | 43.4 | 67.0 | 46.4 |
| MILA-T | MILA | 44M | 255G | 48.8 | 71.0 | 53.6 | 43.8 | 68.0 | 46.8 |
| ConvNeXt-S [33] | CNN | 70M | 348G | 47.9 | 70.0 | 52.7 | 42.9 | 66.9 | 46.2 |
| Swin-S [32] | Transformer | 69M | 354G | 48.2 | 69.8 | 52.8 | 43.2 | 67.0 | 46.1 |
| PVTv2-B3 [44] | Transformer | 65M | 397G | 48.4 | 69.8 | 53.3 | 43.2 | 66.9 | 46.7 |
| FocalNet-S [50] | Transformer | 72M | 365G | 49.3 | 70.7 | 54.2 | 43.8 | 67.9 | 47.4 |
| CSWin-S [9] | Transformer | 54M | 342G | 50.0 | 71.3 | 54.7 | 44.5 | 68.4 | 47.7 |
| Vmamba-S [31] | Mamba | 70M | 384G | 49.9 | 70.9 | 54.7 | 44.2 | 68.2 | 47.7 |
| LocalVMamba-S [25] | Mamba | 69M | 414G | 49.9 | 70.5 | 54.4 | 44.1 | 67.8 | 47.4 |
| MILA-S | MILA | 63M | 319G | 50.5 | 71.8 | 55.2 | 44.9 | 69.1 | 48.2 |

# 6 Conclusion

This paper reveals the surprisingly close relationship between the powerful Mamba and subpar linear attention Transformer, shedding some light on Mamba's superiority and success. We rephrase Mamba as a variant of linear attention Transformer and identify its six major special designs: input gate, forget gate, shortcut, no attention normalization, single-head and modified block design. Empirical validation shows that the forget gate and block design largely enhance performance, while the other distinctions offer marginal contributions or impair model performance. Based on our findings, we propose our Mamba-Inspired Linear Attention (MILA) model by incorporating the merits of these two key designs into linear attention. MILA surpasses various vision Mamba models across multiple tasks, while maintaining parallel computation and high inference speed.

## Acknowledgement

This work is supported in part by the National Natural Science Foundation of China under Grants 42327901 and 62321005.

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

# Appendix

## A    Datasets and Experiment Details

**ImageNet classification.** The ImageNet-1K dataset comprises 1.28 million training images and 50,000 validation images, encompassing 1,000 classes. For a fair comparison, we train our models under the same settings as Swin Transformer [32]. Specifically, we utilize AdamW [34] optimizer to train all our models from scratch for 300 epochs. We apply a cosine learning rate decay schedule with a linear warm-up of 20 epochs and a weight decay of 0.05. The total batch size is 4096 and initial learning rate is set to $4 \times 10^{-3}$. Augmentation and regularization strategies includes RandAugment [6], Mixup [53], CutMix [52], and random erasing [54]. In the training of MILA models, MESA [11] is employed to prevent overfitting.

**COCO object detection.** COCO [30] dataset is a widely adopted benchmark for object detection and instance segmentation with 118K training and 5K validation images. We follow the standard 1x and 3x Mask R-CNN [21] training setting in Swin Transformer [32] to conduct our experiments. The pretrained MILA models are employed as backbones.

**ADE20K semantic segmentation.** ADE20K [55] dataset contains 25K images, 20K for training, 2K for validation, and 3K for testing, with 150 semantic categories. UPerNet [46] is used as the segmentation framework and the same training setting as Swin Transformer [32] is adopted. We report both single-scale and multi-scale testing results.

## B    Additional Experimental Results

**Additional comparison with advanced linear attention designs.** The results are shown in Tab. 6. We empirically find that MILA outperforms various advanced linear attention designs without bells and whistles.

Table 6: Comparison with advanced linear attention designs.

| Method | #Params | FLOPs | Acc. |
|---|---|---|---|
| Hydra Attention [2] | 29M | 4.5G | 80.7 |
| Efficient Attention [40] | 29M | 4.5G | 81.0 |
| FLatten Transformer [15] | 29M | 4.5G | 82.1 |
| SOFT [35] | 24M | 3.3G | 82.2 |
| MILA (Ours) | 25M | 4.2G | 83.5 |

**Ablation on the impact of MESA.** Like in the early stages of studies on visual Transformer, currently vision Mamba research does not have a well-established and universally accepted training protocol. The conventional training setting for vision Transformer may not be optimal for vision Mamba and our Mamba-Inspired Linear Attention. Therefore, in the training of MILA models, we additionally employ the overfitting prevention strategy MESA [11] to fully demonstrate its potential. In Tab. 7, we provide the results without MESA. We can observe that: (1) MESA provides a modest accuracy gain of 0.1-0.3. (2) Without MESA, MILA models still significantly surpass various SOTA vision Mamba models.

## C    Model Architectures

We illustrate the architecture of our MILA model in Fig. 7 and summarize the detailed structure in Tab. 8. We adopt the common 4-stage framework to build MILA model by stacking our MILA blocks at each stage.

## D    Limitations

In this paper, we explore the similarities and disparities between Mamba and linear attention Transformer, providing comprehensive analyses to demystify the key factors behind Mamba's success.

Table 7: MILA models trained without MESA.

| Model | #Params | FLOPs | Acc. |
|---|---|---|---|
| Vim-S [57] | 26M | 5.1G | 80.3 |
| VMamba-T [31] | 31M | 4.9G | 82.5 |
| LocalVMamba-T [25] | 26M | 5.7G | 82.7 |
| MILA-T (w/o MESA) | 25M | 4.2G | 83.3 |
| MILA-T (w/ MESA) | 25M | 4.2G | 83.5 |
| VMamba-S [31] | 50M | 8.7G | 83.6 |
| LocalVMamba-S [25] | 50M | 11.4G | 83.7 |
| MILA-S (w/o MESA) | 43M | 7.3G | 84.2 |
| MILA-S (w/ MESA) | 43M | 7.3G | 84.3 |
| Mamba2D-B [27] | 94M | - | 82.3 |
| VMamba-B [31] | 89M | 15.4G | 83.9 |
| MILA-B (w/o MESA) | 96M | 16.2G | 85.0 |
| MILA-B (w/ MESA) | 96M | 16.2G | 85.3 |

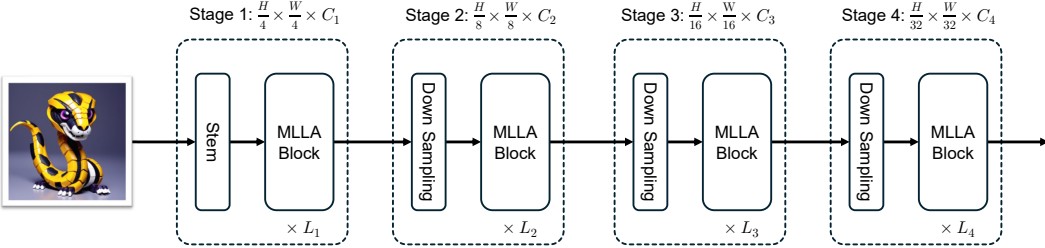

Figure 7: The architecture of MILA model.

Specifically, we begin with the formulas and rephrase Mamba as a variant of linear attention Transformer with six major distinctions: input gate, forget gate, shortcut, no attention normalization, single-head and modified block design. Moreover, we meticulously analyze the pros and cons of each design and prove that the forget gate and block design are the core contributors to Mamba's success. Based on our findings, we propose our Mamba-Inspired Linear Attention (MILA) model, which surpasses various vision Mamba models across multiple tasks, while maintaining parallel computation and high inference speed. However, there may be other small differences between the implementation details of Mamba and linear attention Transformer, and this paper is not exhaustive.

Table 8: Architectures of MILA models.

| stage | output | MILA-T | MILA-S | MILA-B |
|-------|--------|--------|--------|--------|
| res1 | $56 \times 56$ | stem, 64 $\begin{bmatrix} \text{dim 64} \\ \text{head 2} \end{bmatrix} \times 2$ | stem, 64 $\begin{bmatrix} \text{dim 64} \\ \text{head 2} \end{bmatrix} \times 3$ | stem, 96 $\begin{bmatrix} \text{dim 96} \\ \text{head 3} \end{bmatrix} \times 3$ |
| res2 | $28 \times 28$ | downsampling, 128 $\begin{bmatrix} \text{dim 128} \\ \text{head 4} \end{bmatrix} \times 4$ | downsampling, 128 $\begin{bmatrix} \text{dim 128} \\ \text{head 4} \end{bmatrix} \times 6$ | downsampling, 192 $\begin{bmatrix} \text{dim 192} \\ \text{head 6} \end{bmatrix} \times 6$ |
| res3 | $14 \times 14$ | downsampling, 256 $\begin{bmatrix} \text{dim 256} \\ \text{head 8} \end{bmatrix} \times 8$ | downsampling, 256 $\begin{bmatrix} \text{dim 256} \\ \text{head 8} \end{bmatrix} \times 21$ | downsampling, 384 $\begin{bmatrix} \text{dim 384} \\ \text{head 12} \end{bmatrix} \times 21$ |
| res4 | $7 \times 7$ | downsampling, 512 $\begin{bmatrix} \text{dim 512} \\ \text{head 16} \end{bmatrix} \times 4$ | downsampling, 512 $\begin{bmatrix} \text{dim 512} \\ \text{head 16} \end{bmatrix} \times 6$ | downsampling, 768 $\begin{bmatrix} \text{dim 768} \\ \text{head 24} \end{bmatrix} \times 6$ |

