# OpenReview forum: "Demystify Mamba in Vision: A Linear Attention Perspective"
_NeurIPS.cc/2024/Conference — NeurIPS 2024 poster_

### Official Review · Reviewer_XBcf · 2024-07-08

**Soundness:** 3
**Presentation:** 2
**Contribution:** 3
**Rating:** 7
**Confidence:** 5

**Summary:**

This paper presents a thorough analysis of the key factors contributing to the success of the S6 module in Mamba model, and introduces a new linear attention vision network, MLLA, inspired by the S6’s design. Extensive experimental results show outstanding performance, validating the effectiveness of the proposed model.

**Strengths:**

The motivation and analysis underlying this work are fascinating and perceptive, offering a profound examination of the mathematical relationships between Mamba and Linear Attention.

A thorough investigation of the performance contributions of each component from S6 is conducted, with corresponding refinements made to linear attention, ultimately leading to the development of MLLA.

The proposed MLLA serves as a generic vision backbone network, which outperforms recent Mamba-based models, demonstrating its superior capabilities

**Weaknesses:**

I appreciate the paper's insightful analyses, but several key concerns regarding methodology and experiments need to be addressed. If the authors fully resolve these issues, I would be willing to reconsider my evaluation.

**Possible inaccurate analysis of single-head attention**

The analysis of single-head attention appears to be inaccurate and unclear. Generally, multi-head attention in Transformer generates H dynamic matrices during attention computation, where H represents the number of heads. However, the paper seems to interpret S6 as a single-head module without providing sufficient evidence. The authors should explain this point from the perspective of the number of attention matrices generated. Moreover, according to Reference 1, S6 can actually produce multiple dynamic matrices instead of just one.

**Unfair comparison on ImageNet-1K dataset**

The paper uses MESA as its optimizer, improving accuracy compared to AdamW. In contrast, the methods in Table 3 do not use MESA and may also be prone to overfitting. For fairness, authors should ensure that training strategies are consistent with competitors. Moreover, the authors should not hide this apparent difference in the appendix.


**Questionable motivation for using Swin as the baseline**

The motivation for using Swin as a baseline is questionable. One important aspect of linear attention is its ability to efficiently establish global dependencies, so it is unclear why window-based linear attention is used. Additionally, the specific configuration of the baseline model, such as channels and depth, is not provided.

**Unclear setting of d_state in SSM block design**

In the block design, it is unclear what value d_state of SSM is set to.


**Insufficient experiments on downstream tasks**

(1) Semantic segmentation should include tiny, small, and base models. (2) Mask R-CNN 3x should include the base model. (3) The performance comparison of MLLA-T and VMamba-T on object detection is lacking.

**Unfair comparison in Table 6**

The comparison in Table 6 is unfair. The authors aim to compare the performance of different linear attention methods, but different block settings have a significant impact. The authors should use the same architecture, replacing only token mixer module.

**Typo**

In line 294, 'SMM' should be 'SSM'.

**Questions:**

Please see the main weaknesses.

**Limitations:**

Please see the main weaknesses.

---

> ### Author Rebuttal · Authors · 2024-08-02
>
> We would first like to express our appreciation for your time and insightful comments. Please find our response to your concerns in the following:
>
> ---
>
> **1. The analysis of single-head attention.**
>
> Thanks for your insightful comment.
>
> - ***The concept "head" in our paper is the same as its original definition in multi-head attention [1], i.e. the number of groups of $Q, K$, rather than the number of dynamic matrices.*** Since there exists only one set of $C,B$ in S6 (see Eq. 11 of our paper), ***it resembles single-head linear attention.***
> - ***A head can possibly generate multiple dynamic matrices.*** In multi-head attention, one head can only generate a single dynamic matrix $QK^\top$. However, a head of S6 can produce multiple equivalent dynamic matrices. This can be attributed to its special input gate and forget gate design. Specifically, as shown in Eq. 11 and Eq. 12 of our paper, there is only one set of $C\in\mathbb{R}^{N\times d}, B\in\mathbb{R}^{d\times N}$ in S6. Therefore, without considering the input gate $\Delta_i$ and forget gate $\hat{A}_i$, the model can produce only one dynamic attention matrix $CB\in\mathbb{R}^{N\times N}$. However, the input gate $\Delta_i$ and forget gate $\hat{A}_i$ make things different. Specifically, $\Delta_i\in\mathbb{R}^{1\times c}$ and $\hat{A}_i\in\mathbb{R}^{d\times c}$ are not shared across channels $c$, which enables S6 to produce $c$ equivalent dynamic matrices with the single set of $C, B$. Intuitively, S6 generates one attention matrix $CB\in\mathbb{R}^{N\times N}$, yet each channel filters tokens differently based on its input gate weight and introduces unique local biases with its forget gate weight, leading to varied equivalent dynamic matrices.
> - ***In conclusion, S6 uses the single-head design, but its input gate and forget gate lead to multiple equivalent dynamic matrices.***
> - Additionally, we can introduce multi-head design to S6 by initially generating multiple sets of $C^h\in\mathbb{R}^{N\times d}, B^h\in\mathbb{R}^{d\times N},h=1,\cdots,\rm{num\\_heads}$, akin to multi-head attention.
>
> [1] Attention is all you need. In NeurIPS, 2017.
>
> ---
>
> **2. The comparison on ImageNet-1K dataset.**
>
> ***Firstly***, MESA was not utilized in ablation experiments or downstream tasks. These results already fully support the two core findings of this paper: (1) The forget gate and block design are the key designs of Mamba. (2) With the merits of these two key designs, MLLA can outperform various vision Mamba models.
>
> ***Secondly***, we clarify MESA as follows:
>
> - MESA is an overfitting prevention strategy rather than an optimizer.
> - ***Without MESA, MLLA model can also achieve comparable results.*** For example, after removing MESA and increasing drop path rate—a commonly used strategy to prevent overfitting—to 0.3, MLLA-T yields 83.3 accuracy on ImageNet, which still significantly surpasses various SOTA vision Mamba model, e.g. LocalVMamba-T: 82.7, VMamba-T: 82.5, etc.
> - In our early experiments, we found MESA slightly beneficial for MLLA model, but it did not benefit other models as well and even hindered their performance.
>
> ---
>
> **3. The motivation for using Swin as the baseline.**
>
> - ***Window-based attention is not used.*** The shifted window attention in Swin Transformer is replaced with global linear attention to create our baseline model. ***We only employ the macro structure of Swin***, i.e. width, depth and etc., as a clean architecture.
> - The baseline model shares the identical configuration as Swin-T, with widths = [96, 192, 384, 768] and depths = [2, 2, 6, 2].
>
> ---
>
> **4. The setting of d_state in SSM block design.**
>
> The concept of d_state is akin to the head_dim in attention. Therefore, following Swin Transformer, we set it as 32 for all the models in our study.
>
> ---
>
> **5. The experiments on downstream tasks.**
>
> - MLLA's effectiveness on downstream tasks is fully validated by comprehensive COCO object detection and instance segmentation experiments. Hence, we conduct only one experiment on semantic segmentation task to save computation resources.
>
> - As requested, we provide additional results of semantic segmentation task below.
>
>   | Backbone | \#Params | FLOPs | mIoU SS | mIoU MS |
>   | :------: | :------: | :---: | :-----: | :-----: |
>   | VMamba-T |   62M    | 949G  |  47.9   |  48.8   |
>   |  MLLA-T  |   55M    | 932G  |  48.1   |  49.0   |
>
>   Due to time and resource constraints, we couldn't provide the MLLA-S results here.
>
> - To the best of our knowledge, the result for Mask R-CNN 3x + base model is not provided by any vision Mamba models we compared with. Hence, we also omit this experiment to save computation resources.
>
> - Full comparison with VMamba-T on object detection is provided **in the PDF in general response**. MLLA-T achieves comparable results to VMamba-T in the 3x setting but slightly underperforms VMamba-T in 1x training. This can be attributed to MLLA-T having 6M fewer parameters than VMamba-T, possibly requiring longer training to converge. MLLA-S/B significantly outperform VMamba-S/B in both 1x and 3x settings.
>
> ---
>
> **6. The comparison in Table 6.**
>
> - As detailed in our paper, ***the block structure is also a key design of Mamba and our MLLA.*** Hence, we provide a direct comparison with other linear attention methods in Table 6. to validate the effectiveness of our design.
>
> - Here we offer additional comparison under exactly same macro architecture, the Swin-T structure.
>
>   |       Method        | \#Params | FLOPs | Acc. |
>   | :-----------------: | :------: | :---: | :--: |
>   |   Hydra Attention   |   29M    | 4.5G  | 80.7 |
>   | Efficient Attention |   29M    | 4.5G  | 81.0 |
>   | FLatten Transformer |   29M    | 4.5G  | 82.1 |
>   |        MLLA         |   30M    | 4.8G  | 82.6 |
>
>   MLLA shows obviously better result. It is noteworthy that we are unable to provide results for SOFT under this setting since it requires more than 1 hour to train one epoch.
>
> ---
>
> **7. Typo.**
>
> Thanks. It will be corrected.

---

> > ### Comment · Reviewer_XBcf · 2024-08-09
> >
> > I appreciate the authors' response, which has addressed some of my concerns.  Although I understand that the purpose of this paper is to establish connections between various mechanisms in Mamba and Linear Attention, the main issue remains the unfair experimental implementation. The authors should provide results without MESA in Table 3 for tiny, small, and base models, as MESA can boost performance like token labeling, and this should be explicitly indicated. The current presentation may mislead follow-up works due to the unfair training conditions. The authors' claim that other models do not benefit from MESA lacks any supporting evidence.
> >
> > Regarding downstream tasks, while the authors clarify that they did not use MESA, it’s important to note that the pre-trained models have incorporated MESA. This introduces an inherent unfairness in weight initialization, which may affect the fairness of the final performance comparisons.
> >
> > The authors mention that other Mamba-based models have not been tested on 3x schedule-based detection tasks, which raises a question: are Mamba-based models unable to train successfully on 3x schedule-based detection tasks? If so, the potential applications of Mamba-based models would be significantly limited. Additionally, please note that the proposed method belongs to the vision transformer family, not Mamba-based models, and it is crucial to conduct related experiments as listed in many previous works. Reporting results with the 3x schedule is more meaningful, as this allows the models to converge to a more consistent outcome, providing a more reliable comparison.
> >
> > The author should be aware that one of the most crucial aspects of a vision backbone paper is the experimental section, which must adhere to highly consistent training conditions with other competitors to ensure a fair comparison. Implicitly introducing additional tricks to improve performance is misleading and unfair to follow-up works. I must reiterate that the unfair experimental design remains a significant concern.

---

> > > ### Author Response · Authors · 2024-08-09
> > >
> > > We would like to express our appreciation for your comments. However, we believe there are some misunderstandings and we offer clarifications below.
> > >
> > > ---
> > >
> > > **1. MESA does not affect the three main contributions and findings of this work.**
> > >
> > > We would like to emphasize that the three main contributions of our paper remain unaffected by the use of MESA:
> > >
> > > - We reveal Mamba’s close relationship to linear attention Transformer.
> > > - We provide detailed analyses of each special design and validate that the forget gate and block design largely lead to Mamba’s superiority.
> > > - We present MLLA, a novel linear attention model that outperforms vision Mamba models.
> > >
> > > The first finding is thoroughly analyzed in Section 4 of our paper.
> > >
> > > The second one is validated by the results in Table 1 and Table 2, ***where MESA is not employed.***
> > >
> > > The last contribution of our work, MLLA, uses MESA in its training. However, as we already mentioned in our previous response, without MESA, MLLA-T can also achieve 83.3 accuracy and still significantly surpasses various vision Mamba models. Notably, the MLLA model is built to validate our last finding,  i.e. whether linear attention can match or surpass Mamba in vision, rather than to compete against SOTA vision Transformers.
> > >
> > > ***In conclusion, MESA does not influence the core contributions of our work, but rather serves as an additional strategy to help MLLA performs optimally.***
> > >
> > > We further clarify our intent and the reason for using MESA in the following.
> > >
> > > ---
> > >
> > > **2. The reason for using MESA.**
> > >
> > > We would first like to clarify that MESA is just a strategy to prevent overfitting, and it cannot boost performance like token labeling. Token labeling benefits from ***a pre-trained model***, functioning similarly to distillation, whereas MESA only enhances the model's generalization ability and ***doesn't use any pre-trained models.***
> > >
> > > Just like in the early stages of studies on visual Transformer, currently vision Mamba research does not have a well-established and universally accepted training protocol. The conventional training setting for vision Transformer may not be optimal for vision Mamba and our Mamba-Like Linear Attention. Therefore, we additionally employ the overfitting prevention strategy MESA to alleviate the overfitting problem of MLLA model and fully demonstrate its potential. Our goal is to provide the community with more robust models.
> > >
> > > We believe that excessive pursuit of strictly unchanged training setting could actually restrict the exploration of new architectures.
> > >
> > > ---
> > >
> > > **3. Results without MESA.**
> > >
> > > - To better address your request, we further provide the results without MESA.
> > >
> > > - Actually, we already provided the result for ***MLLA-T trained without MESA*** in our previous response. Here, we offer a comparison with vision Mamba models based on this result.
> > >
> > >   |       Model       | #Params | FLOPs | Acc. |
> > >   | :---------------: | :-----: | :---: | :--: |
> > >   |       Vim-S       |   26M   | 5.1G  | 80.3 |
> > >   |     VMamba-T      |   31M   | 4.9G  | 82.5 |
> > >   |   LocalVMamba-T   |   26M   | 5.7G  | 82.7 |
> > >   | MLLA-T (w/o MESA) |   25M   | 4.2G  | 83.3 |
> > >
> > >   ***As you can see, without MESA, MLLA-T can also achieve comparable result and still significantly surpass various SOTA vision Mamba models.***
> > >
> > > - We are currently in the process of training ***MLLA-S/B under the no MESA setting*** and will provide those results ***in a few days.***
> > >
> > > - Furthermore, we will utilize the backbone trained without MESA to conduct downstream tasks.
> > >
> > > - All these results will be included in the revised manuscript. We will provide more experimental results within our capabilities and try our best to benefit the community and follow-up works.
> > >
> > > ---
> > >
> > > **3. Downstream tasks.**
> > >
> > > - There seems to be a misunderstanding. In our previous response, we stated that Mask R-CNN 3x results for ***base level model, e.g. VMamba-B,*** is hardly reported by vision Mamba works. However, we did not claim that Mamba-based models have not been tested on 3x schedule. Indeed, we already provided comparison of ***tiny and small level models under 3x schedule in Table 9. of our paper.***
> > > - The primary focus of our paper is on comparisons with vision Mamba models, rather than achieving SOTA results. Given that the works we compared with do not present 3x detection results for base level model, we also omit this experiment previously. Currently, we are working on this experiment to better address your request.

---

> > > > ### Comment · Reviewer_XBcf · 2024-08-13
> > > >
> > > > I appreciate your clarification. The use of MESA or not does not impact the paper's main contribution. However, I believe it's unfair to directly compare other methods with this work, as it employs additional training techniques. Given the approaching rebuttal deadline, conducting further experiments on small and base models may be difficult. Nevertheless, based on the authors' provided results of MLLA-T (without MESA), I find it acceptable. As a result, I have increased my score. I hope the authors will include comprehensive experimental results in the revised paper.

---

> > > > > ### Author Response · Authors · 2024-08-13
> > > > >
> > > > > Dear Reviewer XBcf, we sincerely appreciate your rigorous research attitude and valuable comments. Here, we provide some additional results we obtained these days to fully resolve your concern.
> > > > >
> > > > > ---
> > > > >
> > > > > **1. Results without MESA.**
> > > > >
> > > > > - ***MLLA models trained without MESA.***
> > > > >
> > > > >   |         Model         | #Params | FLOPs |   Acc.   |
> > > > >   | :-------------------: | :-----: | :---: | :------: |
> > > > >   |         Vim-S         |   26M   | 5.1G  |   80.3   |
> > > > >   |       VMamba-T        |   31M   | 4.9G  |   82.5   |
> > > > >   |     LocalVMamba-T     |   26M   | 5.7G  |   82.7   |
> > > > >   | **MLLA-T (w/o MESA)** |   25M   | 4.2G  | **83.3** |
> > > > >   | **MLLA-T (w/ MESA)**  |   25M   | 4.2G  | **83.5** |
> > > > >   |                       |         |       |          |
> > > > >   |       VMamba-S        |   50M   | 8.7G  |   83.6   |
> > > > >   |     LocalVMamba-S     |   50M   | 11.4G |   83.7   |
> > > > >   | **MLLA-S (w/o MESA)** |   43M   | 7.3G  | **84.2** |
> > > > >   | **MLLA-S (w/ MESA)**  |   43M   | 7.3G  | **84.3** |
> > > > >   |                       |         |       |          |
> > > > >   |       Mamba2D-B       |   94M   |   -   |   82.3   |
> > > > >   |       VMamba-B        |   89M   | 15.4G |   83.9   |
> > > > >   | **MLLA-B (w/o MESA)** |   96M   | 16.2G | **85.0** |
> > > > >   | **MLLA-B (w/ MESA)**  |   96M   | 16.2G | **85.3** |
> > > > >
> > > > >   From these results, we observe:
> > > > >
> > > > >   - The overfitting prevention strategy MESA only provides ***a modest accuracy gain of 0.1-0.3.***
> > > > >   - Without MESA, MLLA models still ***significantly surpass various SOTA vision Mamba models.***
> > > > >
> > > > > - ***Downstream tasks using the backbone trained without MESA.***
> > > > >
> > > > >   |        Method        | Pre-trained Backbone | #Params | FLOPs | $AP^b$ | $AP^m$ |
> > > > >   | :------------------: | :------------------: | :-----: | :---: | :----: | :----: |
> > > > >   | MLLA-T Mask R-CNN 1x |       w/o MESA       |   44M   | 255G  |  46.8  |  42.0  |
> > > > >   | MLLA-T Mask R-CNN 1x |       w/ MESA        |   44M   | 255G  |  46.8  |  42.1  |
> > > > >   | MLLA-T Mask R-CNN 3x |       w/o MESA       |   44M   | 255G  |  48.7  |  43.7  |
> > > > >   | MLLA-T Mask R-CNN 3x |       w/ MESA        |   44M   | 255G  |  48.8  |  43.8  |
> > > > >
> > > > >   The results show that ***the inclusion of MESA during pre-training has a negligible effect on downstream task performance.***
> > > > >
> > > > > - ***We fully appreciate your rigorous research attitude.*** We will include these results and detailed discussions in the revised version.
> > > > >
> > > > > ---
> > > > >
> > > > > **2. Conclusions of the discussion on MESA.**
> > > > >
> > > > > Based on the additional results and our previous discussions, we believe the following conclusions can be made:
> > > > >
> > > > > - ***MESA serves as a strategy to prevent overfitting.***
> > > > > - ***Without MESA, MLLA models can also achieve comparable results in both classification and downstream tasks.*** This is validated by the additional results.
> > > > > - ***MESA does not affect the three main contributions and findings of this work.*** Our work has three main contributions: revealing the relationship between Mamba and linear attention, analyzing and validating the key difference, and presenting MLLA. The first two contributions do not involve MESA. While the last contribution of our work, MLLA, originally used MESA in its training, additional results verify that MLLA models still significantly surpass various SOTA vision Mamba models without it. This further supports our findings. Therefore, MESA does not influence the three core contributions of our work.
> > > > > - ***We employ MESA to provide the community with more robust models.*** Just like in the early stages of studies on visual Transformer, currently vision Mamba research does not have a well-established and universally accepted training protocol. We additionally employ the overfitting prevention strategy MESA to fully demonstrate MLLA's potential. Our goal is to benefit the community and follow-up works. We will include detailed discussions in the revised version.

---

> > > > > > ### Comment · Reviewer_XBcf · 2024-08-13
> > > > > >
> > > > > > Thank you for providing additional results. The paper now offers both insightful methods and solid experimental results, which have contributed to my improved score.

---

### Official Review · Reviewer_dwbt · 2024-07-11

**Soundness:** 4
**Presentation:** 4
**Contribution:** 3
**Rating:** 7
**Confidence:** 3

**Summary:**

The paper takes a deep dive into the workings of Mamba in vision related tasks and compares it to Linear Attention. They conclude that Mamba is a special form of Linear Attention and describe the role that certain parts in the architecture and give them a name that better represent their actual use. They come up with input and forget gate that control the flow of information while shortcuts are closely related to the residual connections. The authors take those learnings and apply them to linear attention  and show that those concepts do improve the linear attention mechanism up to the point where it beats mamba.

**Strengths:**

The authors draw clear connections between Linear Attention and Mamba which is showcased in terms of raw similarity (eq. 11 and 12). Furthermore, they underline this by showing the input gate values on two example images which helps general understanding. They demonstrate the usefulness of the explained modules in Mamba apply them to Linear Attention and show that they do improve performance.

In total a sound paper. The overall gist can be followed very nicely.

**Weaknesses:**

No apparent weaknesses

**Questions:**

- Why did you choose to use Swin Transformer?
- The forget gate creates local bias and position information but also decays the previous hidden state if I understood correctly. When replaced by RoPE or similar the performance increases as shown in Table2. Does this mean that the local bias and the decay can hinder performance?

**Limitations:**

The authors mention the limitations of their work, especially that is it no exhaustive.

---

> ### Author Rebuttal · Authors · 2024-08-03
>
> We would first like to express our appreciation for your time and insightful comments. Please find our response to your concerns in the following:
>
> ---
>
> **1. Reason for using Swin Transformer.**
>
> We offer clarification on employing Swin Transformer:
>
> - Swin Transformer is a widely used architecture in vision tasks. ***We use it as a clean and fair structure*** to validate the effectiveness of each special design of Mamba.
> - Notably, the core operation of Swin Transformer, shifted window attention, is replaced with global linear attention to from our baseline model. Therefore, ***we only employ the macro structure of Swin***—such as width and depth—without employing its window-based attention mechanism.
>
> ---
>
> **2. Replacing the forget gate with positional encoding.**
>
> - Thanks for the insightful question. Yes, the forget gate decays previous hidden state, thus providing local bias and position information.
> - ***Local bias and decay do not hinder performance.*** Instead, they are beneficial for the model. For example, RoPE provides long-term decay and local bias, as demonstrated in its paper [1]. As shown in Table 2 of our study, integrating RoPE into the baseline linear attention model leads to an obvious performance improvement from 77.6 to 80.0. This indicates that local bias and decay can enhance model effectiveness.
> - ***The reason why positional encodings outperform forget gate in Table 2 is that they can enjoy a global receptive field.*** As analyzed in our paper, the forget gate has to employ recurrent calculation, which restricts the receptive field of each token to the preceding sequence, resulting in a causal mode. As verified by a concurrent work, MambaOut [2], such a causal mode can lead to notable performance drop in vision models (see its Fig. 3). Therefore, while the local bias and decay of the forget gate can benefit the model, its causal mode hinders performance. In contrast, positional encodings enable the model to benefit from both local bias and a global receptive field at the same time, thus yielding better results than the forget gate.
>
> [1] Roformer: Enhanced transformer with rotary position embedding. Neurocomputing.
>
> [2] MambaOut: Do We Really Need Mamba for Vision? arXiv:2405.07992.

---

> > ### Comment · Reviewer_dwbt · 2024-08-12
> >
> > Thanks to the authors for their clarifications. I will keep my score unchanged.

---

> ### Author Response · Authors · 2024-08-12
>
> Dear Reviewer dwbt, thank you for your insightful review and for engaging with our work. We would like to know if there are any additional questions or concerns. We are eager to engage in further discussion and provide clarification to fully address them.

---

### Official Review · Reviewer_ii6x · 2024-07-12

**Soundness:** 3
**Presentation:** 3
**Contribution:** 2
**Rating:** 6
**Confidence:** 4

**Summary:**

This paper explores the similarities and differences between the Mamba and linear attention Transformer models. It redefines Mamba as a variant of linear attention Transformer with six key distinctions. The paper also studies each design's impact, identifying the forget gate and block design as key to Mamba's success. Based on these findings, the paper introduces a Mamba-like linear Attention (MLLA) model, which outperforms other vision Mamba models on various tasks while maintaining fast computation and inference speed.

**Strengths:**

* The paper is well-written and well-motivated from the perspective of linear attention.

* The findings are interesting to me.

**Weaknesses:**

* Several works using Mamba for image synthesis should be discussed [1][2][3]. In Line 68, I wonder whether the exploration in this paper is orthogonal to previous explorations on Mamba. For example, Zigma considers layerwise scanpaths with 8 directions. Can the proposed method still yield some improvements on that result?

* Another consideration is that the authors should discuss something with this paper: https://arxiv.org/abs/2406.06484. Check Table 4; it provides a much more general framework that includes most of the linear attention models.

* My third concern is whether the exploration in this work can extend to other linear attention-based models, such as RWKV and xLSTM.

**Questions:**

I have some concerns (see weaknesses) about this paper. I am inclined to raise my score if my concerns are fully resolved.

**Reference:**

[1],Diffusion Models Without Attention.CVPR24.

[2],ZIGMA: A DiT-style Zigzag Mamba Diffusion Model,ECCV24.

[3],Scalable Diffusion Models with State Space Backbone,Arxiv.

**Limitations:**

Yes, they are addressed and discussed.

---

> ### Author Rebuttal · Authors · 2024-08-03
>
> We would first like to express our appreciation for your time and insightful comments. Please find our response to your concerns in the following:
>
> ---
>
> **1. Mamba for image synthesis.**
>
> - Thanks for pointing out these works. ***We will give more credits to them and include detailed discussions in the revised version.***
> - The exploration in this paper is orthogonal to previous explorations on Mamba, and ***our work can further benefit previous works in two ways***:
>   - ***a. Previous studies on Mamba can substitute their SSM with our MLLA block***, which is proved to be more effective and efficient.
>   - ***b. The analyses in our work can help other studies design better Mamba models.*** Our work reveals some suboptimal designs of Mamba, such as the absence of attention normalization, an important component. Hence, previous works on Mamba, e.g. Zigma, can add attention normalization to their model to achieve better results. This enhancement is compatible with other explorations, such as layer-wise scan paths with 8 directions.
> - ***This paper focuses on analyzing.*** Currently we mainly conduct classification, object detection and semantic segmentation experiments, which already validate our analyses and findings. In the future, we will apply MLLA to image synthesis task to develop efficient models.
>
> ---
>
> **2. Discussion with the mentioned paper.**
>
> - Thank you for pointing this out. ***The mentioned paper was made public on June 10, 2024, after the NeurIPS submission deadline of May 22, 2024.***
> - The mentioned work proposes a unified framework for efficient autoregressive sequence transformations, demonstrating various linear attention models within this framework. Despite some similarities with our work, there are fundamental distinctions:
>   - a. Our work not only reveals the close relationship between Mamba and linear attention, but more importantly, ***provides analyses and experiments to verify the effectiveness of each design.*** In contrast, the referenced work mainly develops ***a framework to include*** different linear attention models.
>   - b. ***Our analyses ultimately leads to the development of a novel linear attention model, MLLA***. In contrast, the mentioned work focuses on the development of ***efficient training algorithms for existing models.***
> - ***We will discuss with this paper in the revised manuscript.***
>
> ---
>
> **3. Extension to other linear attention-based models.**
>
> - ***The exploration in this work can be extended to other linear attention-based models.***
>
> - Our work reveals that Mamba can be viewed as a special variant of vanilla linear attention. Other linear attention-based models, e.g. RWKV, xLSTM, can also be viewed as variants of vanilla linear attention. Therefore, the exploration and insights in this work can naturally extend to these models.
>
> - ***We take xLSTM as an example.*** According to Eq. 4 of our paper, linear attention can be written as:
>   $$
>   S_i=S_{i-1}+K_i^\top V_i,\quad  Z_i=Z_{i-1}+K_i^\top, \quad  y_i=Q_i S_i / Q_i Z_i
>   $$
>   Using the same notations, xLSTM can be formulated as:
>   $$
>   S_i=F_i S_{i-1}+I_i K_i^\top V_i,\quad  Z_i=F_i Z_{i-1}+I_i K_i^\top, \quad  y_i=Q_i S_i / \text{max}\\{1, Q_i Z_i\\},
>   $$
>   where $I_i$ is input gate and $F_i$ is forget gate. Additionally, xLSTM employs multi-head design according to its paper. ***Therefore, xLSTM resembles linear attention with additional input gate $I_i$, forget gate $F_i$ and modified normalization $\text{max}\{1, Q_i Z_i\}$.*** As a result, many explorations of this work can be extended to xLSTM. For instance, we can make ***bold speculations*** like:
>
>   - a. The forget gate $F_i$ is likely to play a crucial role in xLSTM.
>   - b. When applying xLSTM to vision models, the forget gate could possibly be replaced with positional encodings. Just like what we did in Table 2.
>   - c. The input gate $I_i$ may not offer significant benefits in vision tasks.
>   - d. Enhanced block designs from Mamba could potentially benefit xLSTM.
>   - e. The modified normalization $\text{max}\\{1, Q_i Z_i\\}$ might provide greater stability compared to vanilla linear attention, warranting further exploration.
>
> - ***In conclusion, the analyses and insights of our work can extend to other linear attention-based models, helping them improve interpretability, further enhance models, etc.***

---

> > ### Comment · Reviewer_ii6x · 2024-08-09
> >
> > Thank you for your response. After reading the response and other reviewers' comments, I believe that my concerns are mostly resolved and I will increase my rating from 4 to 6.

---

> > > ### Author Response · Authors · 2024-08-10
> > >
> > > Thank you again for your time and valuable comments.

---

### Official Review · Reviewer_QFdu · 2024-07-13

**Soundness:** 3
**Presentation:** 3
**Contribution:** 3
**Rating:** 7
**Confidence:** 5

**Summary:**

This paper primarily discusses the similarities and differences between the Mamba model and the Linear Attention Transformer, and conducts an in-depth analysis of the key factors contributing to Mamba's success in visual tasks. The paper elaborates on six major design differences of Mamba compared to the Linear Attention Transformer: input gate, forget gate, shortcut connection, attention-free normalization, single-head attention, and modified block design. For each of these designs, the author meticulously analyzes their advantages and disadvantages, and evaluates their applicability in visual tasks.

Through experimental research, the paper emphasizes that the forget gate and block design are the core contributors to Mamba's success. Based on these findings, the author proposes a model called Mamba-Like Linear Attention (MLLA), which integrates the advantages of the forget gate and block design into the linear attention framework. Experimental results show that MLLA outperforms various visual Mamba models in image classification and high-resolution dense prediction tasks, while maintaining parallel computing capabilities and fast inference speed.

**Strengths:**

1. The paper provides a new perspective for understanding Mamba's success by thoroughly analyzing the similarities and differences between the Mamba model and the Linear Attention Transformer. By rephrasing formulas, the paper considers Mamba as a variant of the Linear Attention Transformer and clearly points out Mamba's six design features, including input gate, forget gate, shortcut connection, attention-free normalization, single-head attention, and modified block design.

2. Each design aspect is meticulously analyzed, and its advantages and disadvantages in visual tasks are evaluated. Experimental results verify that the forget gate and block design significantly contribute to Mamba's performance improvement, while other designs either have minimal marginal contributions or may potentially harm model performance.

3.  The Mamba-Like Linear Attention (MLLA) model is proposed, combining the advantages of the forget gate and block design in linear attention. The MLLA model outperforms existing Mamba models in image classification and high-resolution dense prediction tasks while maintaining computational parallelization and high-speed inference capabilities.

**Weaknesses:**

There are some minor issues:

1. In the final MLLA model, there is no SSM but a forget gate. The forget gate was originally proposed in RNN and LSTM. Thus calling it a Mamba-like model seems to be improper.

2. Previous "Demystify" papers are not cited in the paper, e.g., [a, b, c]. The final architecture MLLA follows the macro structure of Swin and [a,b,c] all show that the macro structure is very important but the local blocks are not so important. The paper should build a connection with the previous "Demystify" papers and tell us what are the most effective parts?

[a] On the Connection between Local Attention and Dynamic Depth-wise Convolution, ICLR 2022
[b] What Makes for Hierarchical Vision Transformer? DOI: 10.1109/TPAMI.2023.3282019
[c] Demystify Transformers & Convolutions in Modern Image Deep Networks, arXiv:2211.05781

**Questions:**

1. How many GPUs are used to perform this research, including the training cost and ablation experiments cost?
2. Why Fig. 1 is placed on page 2 but cited on page 4?
3. What are the results of Mamba + Swin?

**Limitations:**

Nope.

---

> ### Author Rebuttal · Authors · 2024-08-03
>
> We would first like to express our appreciation for your time and insightful comments. Please find our response to your concerns in the following:
>
> ---
>
> **1. The name of MLLA model.**
>
> Thanks for your comment.
>
> ***Firstly***, we name the final model Mamba-Like Linear Attention because it incorporates two key designs from Mamba: the forget gate and block design. Given that there is no SSM in the final model, maybe Mamba-Inspired Linear Attention (MILA) is a better name?
>
> ***Secondly***, we are very happy to rename the model if a more suitable name is suggested.
>
> ---
>
> **2. Previous "Demystify" papers.**
>
> Thanks for the valuable suggestion. We offer discussions with these works.
>
> [1] studies the connection between local attention and dynamic depth-wise convolution, validating that dynamic depth-wise convolution can perform on-par with or slightly better than local window attention. It highlights the effectiveness of larger kernel sizes and input-dependent weights for vision tasks. [2] replaces the attention operations in Swin model families with simple linear mapping layers and shows that the macro architecture may be more responsible for high model performance. [3] proposes a unified macro architecture to identify the real gains of popular convolution and attention operators. It demonstrates that different token mixers achieves varying performance under the same macro architecture, with Halo attention and deformable convolution yield optimal results.
>
> Based on these papers and our findings regarding the forget gate and block design, we boldly make the following inferences:
>
> - ***Both macro and micro structures are important***.  [1, 2] emphasize the importance of macro structure, while [3] shows that superior token mixers can enhance performance under the same macro architecture. In our study, the forget gate and block design serve as micro and macro structures, respectively, both playing significant roles.
> - ***Input-dependent feature, large receptive field and local bias are beneficial for effective micro token mixer designs.*** [1, 3] and our paper suggest these properties are effective. This further explains the effectiveness of MLLA, which incorporates all these features while maintaining linear complexity.
>
> **We will cite these related works and provide detailed discussion in the revised version.**
>
> [1] On the Connection between Local Attention and Dynamic Depth-wise Convolution, ICLR 2022.
>
> [2] What Makes for Hierarchical Vision Transformer? DOI: 10.1109/TPAMI. 2023.3282019.
>
> [3] Demystify Transformers & Convolutions in Modern Image Deep Networks, arXiv:2211.05781.
>
> ---
>
> **3. Training cost.**
>
> 32 GPUs are used to perform this research. Each tiny-scale model, including ablation experiments, requires around 12 hours for training on ImageNet-1K with 32 GPUs.
>
> ---
>
> **4. The position of Fig. 1.**
>
> Thanks for your question. Fig. 1 is placed on page 2 as it serves as the main figure of our paper. Considering it is cited on page 4, we will adjust its position.
>
> ---
>
> **5. The results of Mamba + Swin.**
>
> Due to time and computation resource constrains, we are unable to provide the results of Mamba + Swin architecture. However, we find that the initial version of VMamba yields pertinent findings by applying Mamba to Swin architecture in a cross-scan approach. The results are provided below.
>
> |                Model                 | #Param | FLOPs | Acc  |
> | :----------------------------------: | :----: | :---: | :--: |
> | Mamba+Swin-T (implemented by VMamba) |  22M   | 4.5G  | 82.2 |
> |                MLLA-T                |  25M   | 4.2G  | 83.5 |

---

> ### Author Response · Authors · 2024-08-12
>
> Dear Reviewer QFdu, thank you for your insightful review and for engaging with our work. We would like to know if there are any additional questions or concerns. We are eager to engage in further discussion and provide clarification to fully address them.

---

### Author Rebuttal · Authors · 2024-08-03

We thank all the reviewers for their insightful and valuable comments.

We have carefully considered the reviewers' comments and provided additional clarification to address each concern. Here, we offer general responses to all reviewers on two key issues.

---

**1. The motivation of our work.**

This work primarily focuses on ***demystifying the key factors behind Mamba’s success***, rather than developing a series of models to achieve SOTA results. ***Our analyses and verifications form the core of this work***, with MLLA models serving to validate whether the subpar linear attention can match or even surpass the high-performance of Mamba.

---

**2. Discussion with related works.**

Due to recent abundance of Mamba-related works, we may have overlooked some important related works. We appreciate the reviewers for highlighting these works. We will give more credits to them and provide detailed discussions in the revised manuscript.

---

**For detailed responses to individual reviewer comments, please refer to our separate responses to each reviewer.**

Lastly, we would like to thank the reviewers for their time and we are welcome for any further discussion.

---

### Decision · Program_Chairs · 2024-09-25

**Decision:**

Accept (poster)

**Comment:**

The manuscript has been reviewed by several reviewers, all of whom, after rebuttal, gave positive ratings and agreed that the manuscript meets the bar of NeurIPS.

Essentially, the reviewers were happy to find that the study is well motivated and the design is well conducted. There are a few flaws but the reviewers are convinced after the rebuttal.

The AC agrees with the consensus from the reviewers and therefore commends the manuscript to be accepted. Congratulations!